# PPARγ Acetylation in Adipocytes Exacerbates BAT Whitening and Worsens Age-Associated Metabolic Dysfunction

**DOI:** 10.3390/cells12101424

**Published:** 2023-05-18

**Authors:** Ying He, Ruotong Zhang, Lexiang Yu, Tarik Zahr, Xueming Li, Tae-Wan Kim, Li Qiang

**Affiliations:** 1Naomi Berrie Diabetes Center, Columbia University, New York, NY 10032, USA; 2Department of Pathology and Cell Biology, Columbia University, New York, NY 10032, USA; twk16@cumc.columbia.edu; 3Department of Molecular Pharmacology and Therapeutics, Columbia University, New York, NY 10032, USA; 4Stuyvesant High School, New York, NY 10032, USA; 5Taub Institute of Research on Alzheimer’s Disease and the Aging Brain, Columbia University, New York, NY 10032, USA

**Keywords:** PPARγ acetylation, brown adipose tissue, whitening, metabolic dysfunction

## Abstract

Aging and obesity are the two prominent driving forces of metabolic dysfunction, yet the common underlying mechanisms remain elusive. PPARγ, a central metabolic regulator and primary drug target combatting insulin resistance, is hyperacetylated in both aging and obesity. By employing a unique adipocyte-specific PPARγ acetylation-mimetic mutant knock-in mouse model, namely aKQ, we demonstrate that these mice develop worsened obesity, insulin resistance, dyslipidemia, and glucose intolerance as they age, and these metabolic deregulations are resistant to intervention by intermittent fasting. Interestingly, aKQ mice show a whitening phenotype of brown adipose tissue (BAT) manifested in lipid filling and suppressed BAT markers. Diet-induced obese aKQ mice retain an expected response to thiazolidinedione (TZD) treatment, while BAT function remains impaired. This BAT whitening phenotype persists even with the activation of SirT1 through resveratrol treatment. Moreover, the adverse effect of TZDs on bone loss is exacerbated in aKQ mice and is potentially mediated by their increased Adipsin levels. Our results collectively suggest pathogenic implications of adipocyte PPARγ acetylation, contributing to metabolic dysfunction in aging and thus posing as a potential therapeutic target.

## 1. Introduction

Aging and obesity are the two primary driving forces of chronic health complications such as diabetes, cardiovascular disease, and cancer [1]. These two manifestations overlap on escalating the dysregulation of metabolic pathways, while also mutually promoting one other [2]. Peroxisome-proliferator-activated receptor gamma (PPARγ) has become increasingly appreciated as a knot connecting aging and obesity. It plays vital roles in regulating glucose and lipid metabolism, adipocyte biology, insulin sensitivity, and the inflammatory response, making it closely involved in the pathophysiology and intervention of obesity and its comorbidities. Its synthetic thiazolidinedione (TZD) agonists, such as rosiglitazone and pioglitazone, are the most potent insulin sensitizers; however, various side effects are common. 

Emerging evidence implicates PPARγ in the aging process. The PPARγ2 isoform polymorphism Pro12Ala, for example, is associated with longevity in humans [3], and mouse models conferring PPARγ2 deficiency have exhibited a reduced lifespan [4,5]. A clinical study found that the expression of PPARγ is inversely correlated with age and negatively correlated with the presence of reactive oxygen species, total free fatty acids, and palmitic acid in the human liver [6]. Interestingly, interventions known to extend lifespan, such as a moderately high-fat diet (IHF), are accompanied by PPARγ upregulation [6]. Moreover, the treatment of aged mice with a TZD agonist prolongs lifespan and alleviates the associated metabolic hurdles [7]. As such, PPARγ has been implicated as a determinant of longevity [5]; however, how PPARγ is dysregulated in aging and how that ties into its deregulation in obesity remains poorly understood. 

PPARγ is a nuclear receptor that is activated upon ligand binding to heterodimerize with RXRα, forming a complex suitable for DNA binding [8]. It is also regulated at the transcriptional level to sense nutrient availability, and its expression is activated by insulin [9,10,11]. Beyond these two primary regulatory layers, PPARγ undergoes various posttranslational modifications (PTMs), including acetylation, phosphorylation, and SUMOylation [12,13,14,15]. It can be appreciated that PTMs provide a new perspective on PPARγ function in transcriptional selectivity. Previous studies have shown that SirT1 deacetylates PPARγ in a ligand-dependent manner to induce brown remodeling of white adipose tissue (WAT) by selectively activating catabolic genes, without affecting canonic downstream targets [16]. Indeed, PPARγ undergoes pronounced acetylation in obesity and aging [17]. Mice carrying K268R/K293R mutations, mimicking constitutive deacetylation (the 2KR model), are protected from obesity and, more importantly, resistant to TZD’s adverse effects, while retaining the insulin sensitization response [18]. Given that the 2KR mouse is a whole-body knock-in model, we recently developed an adipocyte-specific PPARγ acetylation-mimetic mutant (K293Q) knock-in model (*Pparγ^KQ/KQ^*: *Adipoq-Cre*, aKQ) to dissect the contribution of adipocyte PPARγ. Consequently, PPARγ acetylation in adipocytes is important for adipose adaption to nutrient challenges, as well as for the orchestration of metabolic rhythms [17]. 

PPARγ is hyperacetylated in aging [17], and consistently, the metabolic protections of PPARγ deacetylation are exaggerated in aged 2KR mice, such as the inhibition of visceral adiposity [18] and protection against atherosclerosis [19]. Therefore, in this present study, we evaluated the hypothesis that PPARγ acetylation contributes to aging-associated adipose dysfunction using our unique aKQ mouse model. We further investigated their response to TZD treatment, including assessing the adverse effects on the bone. aKQ mice presented with impaired insulin sensitivity in aging and worsened TZD-induced bone loss. Intriguingly, these detriments were accompanied by an accelerated whitening of brown adipose tissue (BAT). Collectively, our study demonstrates PPARγ acetylation in adipose tissue as a pathogenic factor underlying metabolic dysfunction in aging and obesity. 

## 2. Materials and Methods

### 2.1. Animal Studies

aKQ mice on a C57BL/6J background were generated as described previously [17]. Mice were housed at room temperature (RT, 23 ± 1 °C) in a 12 h light/dark cycle (07:00 a.m./19:00 p.m.) barrier facility. The 60% high-fat diet (HFD) was purchased from Research Diets, Inc. (New Brunswick, NJ, USA, D12492i). Rosiglitazone maleate (Abcam, ab142461) was mixed in the 60% HFD at a dose of 100 mg/kg by Research Diets (New Brunswick, NJ, USA). Our intermittent fasting method adopted an every-other-day regimen by removing food with free access to water 1 h before the lights turned off (19:00 p.m.) for 24 h and adding food back at the same time on the next day [20]. This process was repeated for 6 weeks in 1-year-old male aKQ mice and their littermate controls.

For insulin tolerance tests (ITT), the mice were fasted for 5 h in cages with fresh bedding. Body weight (BW) and fasting glucose were recorded, the mice were intraperitoneally (i.p.) injected with insulin (0.75 U insulin kg^−1^ BW), and glucose was measured using a OneTouch Ultra glucometer at 15, 30, 45, and 60 min. For the glucose tolerance tests (GTT), the mice were fasted for 16 h. After recording BW and fasting glucose, the mice were injected with glucose (i.p., 2 g kg^−1^ BW). Blood glucose was measured at 15, 30, 60, 90, and 120 min after injection. The body compositions were determined using an EchoMRI (EchoMRI LLC, Houston, TX, USA). Plasma insulin, non-esterified fatty acids (NEFA), and triglycerides (TG) were determined with the following kits: mouse insulin ELISA (Mercodia, Uppsala, Sweden), NEFA-HR (Thermo Fisher Scientific, Waltham, MA, USA), and Infinity Triglyceride Reagent (Thermo Fisher Scientific, Waltham, MA, USA), respectively.

### 2.2. Western Blotting 

Tissues were homogenized with a Polytron homogenizer in an IntactProein extraction buffer (GenuIn Biotech, Blacksburg, VA, USA, catalog #415). After placing on ice for 15 min, the lysates were sonicated for 10 min, and debris was removed by centrifugation at 14,000 rpm for 10 min at 4 °C. The total protein was resolved on 8% or 10% SDS-PAGE after BCA determination of the protein concentration. The Western blotting signals were detected with ECL (Thermo Fisher Scientific, Waltham, MA, USA, catalog PI32209). The following antibodies were used in this study: anti-HSP90 (Proteintech, Rosemont, IL, USA, catalog # 13171-1-AP), anti-Adipsin (R&D systems, Minneapolis, MN, USA, catalog # AF5430), anti-Adiponectin (Invitrogen, Waltham, MA, USA, catalog # PA1-054), anti-PPARγ (Cell Signaling Technology, Danvers, MA, USA, catalog # 2443), anti-SirT1, anti-UCP1 (Abcam, Cambridge, UK, ab234430), anti-PGC-1α (Abcam, Cambridge, UK, catalog # ab54481), anti-C/EBPα (Santa Cruz, CA, USA, catalog # sc-61). Western blots were quantified using densitometry via Image J.

### 2.3. Bone Processing and Analysis 

The femurs were collected after removing proximal muscle and connective tissue were and fixed in a 10% neutral buffered formalin solution overnight at 4 °C. For each mouse, one femur was used for bone microarchitecture analysis and lipid quantification using a Quantum FX μCT Scanner (PerkinElmer, Waltham, MA, USA), and the other femur was used to extract bone marrow for RNA analysis. For lipid determination, the femurs were decalcified for at least two weeks in a 14% EDTA solution, which was changed every three days. The bones were then stained in a 1% osmium tetroxide and 2.5% potassium dichromate solution at room temperature for 48 h and washed in tap water for at least 2 h. The bones were imaged by μCT. Bone mineral density (BMD) and lipid volume were quantified using the software Analyze 12.0 [21]. 

### 2.4. Gene Expression 

RNA from tissues (including bone marrow) were isolated using a TRI-Isolate RNA Pure kit (IBI Scientific, Dubuque, IA, USA). Then, 1 μg total RNA was used for reverse transcription to cDNA using the High-Capacity Reverse Transcription Kit (Applied Biosystems, Waltham, MA, USA). Quantitative Real-Time PCR (qPCR) was conducted on a Bio-Rad CFX96 Real-Time PCR system with an AzuraView GreenFast qPCR Blue Mix LR (Azura Genomics, Waltham, MA, USA). The relative gene expression levels were calculated using the ∆∆Ct method with Cyclophilin A (CPA) as the reference gene.

### 2.5. Hematoxylin and Eosin (H & E) Staining

Fresh tissues from mice were immediately fixed in a 10% neutral buffered formalin solution for 24 h and then dehydrated in 70% ethanol at 4 °C for another two days. The samples were embedded into the paraffin and cut into 5 μm sections. Allocated sections were stained with hematoxylin and then washed with water three times. After a step of 1% (*v*/*v*) hydrochloric acid alcohol differentiation, the sections were washed with water three times and then stained with eosin. Following dehydration and mounting, the images were photographed using a microscope (Olympus I X 71) with a DP74 camera. 

### 2.6. Immunostaining of Ucp1

Mice BATs were fixed in a 10% formalin solution for 16 h, followed by storage in 70% ethanol for subsequent paraffin embedding. Next, sections of 6 μm in thickness were obtained onto charged slides. The slides were hydrated in Xylenes and descending concentrations of ethanol. Using a pressure cooker, slides underwent heat-induced antigen retrieval in a 10 mM sodium citrate solution, were cooled on ice, and washed with 1 × PBS prior to treatment with 3% H_2_O_2_ for 10 min at room temperature to prevent background staining. The slides were then washed and blocked in a blocking solution (1 × PBS, 0.1% Tween-20, and 5% normal goat serum) for 1 h before incubating with a primary antibody against UCP-1 (1:50, Cell Signaling Technology, Danvers, MA, USA, catalog #72298) for 16 h at 4 °C. An HRP-conjugated secondary antibody was used the next day (1:200, Millipore Sigma, Burlington, MA, USA, catalog #A0545) for 1 h at room temperature and developed using the ImmPACT DAB Substrate Kit (Vector Laboratories, Newark, CA, USA, catalog SK-4105). Dehydration was performed using ascending concentrations of ethanol, ending in Xylenes. Slides were mounted with coverslips using Permount Mounting Medium (Thermo Fisher Scientific, Waltham, MA, USA, SP15-500) and imaged under light microscopy using a Keyence imaging instrument.

### 2.7. Statistics 

To evaluate statistical significance, we conducted two-way ANOVA or two-tailed Student’s *t*-tests using the GraphPad Prism 6.0 software (GraphPad Software). A value of *p* < 0.05 was considered statistically significant. Quantitative data are expressed as mean ± SEM (standard error of the mean). 

### 2.8. Study Approval

All of the studies in animals were approved by the Columbia University Animal Care and Utilization Committee. 

## 3. Results

### 3.1. PPARγ Acetylation in Adipocytes Exacerbates the Age-Associated Metabolic Decline

In a previous report, we found constitutive PPARγ acetylation in adipocytes (aKQ model) to impair adipose plasticity in response to nutrient challenges [17]. These aKQ mice displayed normal body weight at 8 weeks old, but interestingly gained more body weight than their controls at 24 weeks old (Figure 1A). This increased body weight was ascribed to higher fat mass but not lean mass (Figure 1B,C), owing to the decreased energy expenditure previously observed [17], while the aKQ mice had comparable food intake to the WT mice (Figure 1D). This pro-obesity phenotype is age-associated, becoming more significant at 1 year old [17]. Given their augmented adiposity, middle-aged aKQ mice displayed worsened insulin sensitivity and glucose tolerance (Figure 1E,F). In line with their insulin resistance, they had increased plasma NEFA levels, while their plasma TG levels were comparable to that of the control mice (Figure 1G,H). Moreover, plasma adipsin levels were increased in middle-aged aKQ mice (Figure 1I,J), reinforcing that Adipsin is a specific downstream target of PPARγ acetylation [18]. In contrast, their adiponectin levels remained constant (Figure 1I,J). Collectively, these data demonstrate that PPARγ acetylation in adipocytes intensified the metabolic complications associated with aging, consistent with observed hyperacetylation of PPARγ observed in aging [17].

### 3.2. PPARγ Acetylation Promotes Lipid Accumulation in BAT during Aging

Upon sacrifice, there was no increase in the mass of inguinal white adipose tissue (iWAT) and epididymal white adipose tissue (eWAT) in middle-aged aKQ mice, despite their modestly higher body fat composition nor liver size (Figure 2A). The morphology of iWAT and eWAT appeared relatively normal in terms of adipocyte size (Figure 2B,C); however, more lipid-filling brown adipocytes, some of which were even unilocular, were observed in the BAT of aKQ mice (Figure 2B). Lipid accumulation in BAT to achieve a WAT-like morphology is called “whitening”, a process often seen in aging as an indicator of BAT degeneration [22]. In BAT, there was a modest increase in Adipsin, but a significant decrease in Adiponectin in aKQ mice (Figure 2D,E). We confirmed efficient replacement of WT PPARγ (K293K) in BAT with mutant K293Q (Figure 2F). Moreover, aKQ modestly repressed the expression of the browning marker *Ucp1*, and increased the expression of the whitening marker *Adipsin* and proinflammatory cytokine *Il6*, with no changes in the gene expression of *Adipoq*, *Fabp4*, and *Cpt1a* (Figure 2F). UCP1 protein levels were not altered in BAT (Figure 2G and Appendix A). A similar trend in Adipsin was observed in iWAT, although PPARγ was repressed (Figure 2H,I). Consistently, browning markers *Ucp1*, *Dio2*, and *Cidea* were not altered in eWAT, hinting no negative effects on WAT brown remodeling at the basal condition in middle-aged aKQ mice (Appendix A). Therefore, PPARγ acetylation shows a particular effect on accelerating BAT whitening during aging. This finding is consistent with the persistent inhibition of lipid accumulation in BAT of PPARγ deacetylation-mimetic 2KR mice [18]. 

### 3.3. Metabolic Improvements of Intermittent Fasting (IF) Are Dampened in Middle-Aged aKQ Mice

Intermittent fasting (IF) is emerging as a potent intervention against metabolic dysfunction, particularly in obesity and aging [23,24]. The every-other-day method of IF has been demonstrated to reduce fat body composition, improve glucose tolerance, and decrease frailty in middle-aged male mice [20]. As expected, 6 weeks of IF reduced the total body weight in both the controls and aKQ middle-aged mice (1 year old) (Figure 3A), with a blunting of the latter’s higher fat content (Figure 3B). Despite their same body composition (Figure 3B,C), aKQ mice remained less sensitive to insulin stimulation (Figure 3D), and their tolerance to glucose was consistently worse (Figure 3E). Moreover, their plasma NEFA levels were no longer higher than the control mice after IF (Figure 3F); instead, their TG levels were higher (Figure 3G). The increase in circulating adipsin levels persisted in aKQ mice after 6 weeks of IF, accompanied by an increased expression in WAT (Figure 3H,I). Of note, the BAT whitening phenotype in aKQ mice remained exacerbated in IF (Figure 3J), further supported by impaired brown adipocyte gene expression (*Ucp1*) and augmented whitening markers (*Adipsin* and *Leptin*) and the key lipogenic gene *Fasn* (Figure 3K), even though there was no significant decrease in UCP1 protein expression (Appendix A). Moreover, aKQ appeared to have an impairment in lipolysis (Appendix A). However, the overall phenotype in the WATs of aKQ mice on IF is inconspicuous in terms of adipocyte hypertrophy and gene expression (Appendix A). Additionally, it is worth mentioning that there is increased lipid accumulation in BAT upon IF (Appendix A), in line with a mild increase in BAT mass after IF [25]. This IF-induced whitening phenotype was also reported in rats [26], indicating that BAT has a distinct response to IF compared with WAT. Therefore, deacetylation of PPARγ is required to fully execute the metabolic improvements of IF.

### 3.4. aKQ Mice Retain the Response to TZDs in Diet-Induced Obesity but with Impaired BAT Function

TZDs are full agonists of PPARγ that potently improve insulin sensitivity [27,28]. Next, we asked whether PPARγ acetylation in adipocytes affects the response to TZD treatment. To test this, we induced insulin resistance in the control and aKQ mice by feeding them an HFD for 16 weeks, with subsequent treatment of rosiglitazone. aKQ mice displayed a similar body weight and composition as control mice after rosiglitazone treatment (Figure 4A). aKQ mice had comparable insulin sensitivity (Figure 4B) and glucose tolerance (Figure 4C) to WT mice, no matter before or after rosiglitazone treatment, showing aKQ mice are still responsive to rosiglitazone (Figure 4D,E). Adipsin levels in the circulation were persistently increased in aKQ mice, while the adiponectin levels remained constant (Figure 4F,G). Upon sacrifice, there was no difference in iWAT, eWAT, and liver mass (Figure 4H). Given the blunted metabolic phenotype in aKQ mice, we did not focus on white adipose tissue. Instead, we further examined BAT since TZDs are well known to induce lipid filling in BAT [16]. Interestingly, this “whitening” phenotype of BAT was magnified in aKQ mice. The brown adipocyte marker UCP1 was decreased in the BAT of aKQ mice, together with the deacetylase of PPARγ, SirT1, which is enriched in BAT and conveys a pro-BAT function (Figure 4I,J) [16]. Adipsin, as a white adipocyte marker, was increased, whereas Adiponectin and its upstream regulator C/EBPα were both repressed. Therefore, despite retaining the full insulin-sensitizing response to TZD, aKQ mice remained impaired in BAT function. 

### 3.5. Activation of SirT1 Fails to Protect aKQ Mice from BAT Whitening

TZD-induced PPARγ deacetylation is SirT1 dependent [16]. Our recent work revealed significant elevation in PPARγ acetylation levels in aging and obesity, in which SirT1 activity was diminished [17]. The lack of phenotype in the WAT of DIO aKQ mice can likely be explained by the K293Q mutation mimicking the hyperacetylation of wildtype PPARγ under low SirT1 activity. We then reasoned that the BAT whitening phenotype in aKQ mice should be attenuated by SirT1 activation if it is not dependent on PPARγ acetylation. To test this, we treated DIO WT and aKQ mice with resveratrol to activate SirT1 [29], while maintained rosiglitazone treatment to ensure full agonism of PPARγ. Three weeks of combined treatment of resveratrol and rosiglitazone did not alter body weight changes in aKQ mice after obesity induction, but caused slightly worse insulin sensitivity and more favorable glucose tolerance in aKQ mice (Figure 5A–C). Despite these minor metabolic changes, there was clearly more lipid accumulation in the BAT of aKQ mice compared with control groups using H & E staining (Figure 5D). The WATs of aKQ mice were barely affected in terms of depot size and morphology (Figure 5D,E). The browning markers UCP1 and PGC-1α were consistently repressed in aKQ mice; however, the expression of Adiponectin and Adipsin was unchanged despite the upregulation of C/EBPα (Figure 5F,G). The decreased UCP1 was further confirmed by immunohistochemical staining (Figure 5H,I). These results suggest that PPARγ acetylation directly causes whitening of BAT by promoting lipid filling and dampening thermogenic activity. 

### 3.6. PPARγ Acetylation in Adipocytes Worsens TZD-Induced Bone Loss

Bone loss is the most prevalent adverse effect of TZD treatment and can be inhibited by PPARγ deacetylation using a whole-body knock-in model of a constitutive deacetylation-mimetic 2KR mutant [18]. This bone loss effect involves active remodeling of both adipogenic and osteogenic programs. Here, we used the aKQ model to specifically dissect the contribution of PPARγ acetylation in adipocytes to this drawback. Three weeks of rosiglitazone treatment in the presence of resveratrol resulted in a lower bone mineral density (BMD) in the trabecular region of aKQ mice, without affecting bone volume (Figure 6A,B). The cortical region was not affected (Figure 6C,D). Osmium tetroxide staining of lipid droplets revealed a more than two-fold increase in bone marrow adiposity in aKQ mice, although not significant (Figure 6E,F). In line with their lower BMD, aKQ inhibited the expression of key osteoblast markers *Alp* and *Runx2* in the bone marrow (Figure 6G), indicating an impaired osteogenic gene expression. Hence, PPARγ acetylation in adipocytes worsens TZD-induced bone loss.

## 4. Discussion

Aging and obesity are the two prominent risk factors contributing to metabolic dysfunction [30], and both are associated with the degeneration of BAT [22,31]. Interestingly, PPARγ acetylation is increased in both conditions [17]. Using an adipocyte-specific PPARγ acetylation-mimetic mouse model (aKQ), we demonstrate a BAT whitening phenotype accompanied by impaired insulin sensitivity and disordered lipid and glucose metabolism in aging. Interestingly, despite the profound benefits of IF that have been reported regarding insulin sensitivity, oxidative stress, circadian rhythm, and longevity [24,32,33,34,35], IF failed to fully abrogate the metabolic detriments in aKQ mice. Therefore, PPARγ acetylation specifically on Lys293 in adipocytes is a pathogenic factor predisposing mice to metabolic complications in aging. This is probably due to impairing adipose plasticity, which has been embodied in young aKQ mice with nutrient challenges of HFD and calorie restriction [17]. In this regard, PPARγ acetylation on the Lys293 residue could serve as an indicator of metabolic health and thus be a therapeutic target. It is conceivable that compounds that induce deacetylation on this residue, probably even without a PPARγ agonist activity, might be used to curb metabolic decline during aging. 

BAT is an energy-dissipating organ, naturally counteracting the positive energy balance in obesity [36]. It is highly vascularized and contains dispersed lipid droplets, in sharp contrast with the unilocular lipid droplet in white adipocytes. However, this typical brown adipocyte morphology is switched to white adipocyte-like when BAT function is compromised, such as in obesity [31], aging [22], thermoneutrality [37], and diabetes [38]. This whitening process is characterized by lipid filling and suppressed BAT thermogenic function. Interestingly, both phenotypical changes are persistently observed in aKQ mice under multiple conditions, including aging, IF, and rosiglitazone treatment, regardless of the activation of SirT1. This phenotype in aKQ mice is in line with the inhibited BAT whitening observed in the deacetylation-mimetic PPARγ mutant knockin 2KR mice [18]. Therefore, PPARγ acetylation is a possible sensor and modulator of whitening, mirroring the browning function of PPARγ deacetylation. The detailed molecular mechanisms of tuning PPARγ acetylation and the consequent transcriptional regulation of downstream target genes warrant further investigation. 

TZD’s clinical utilizations have been greatly limited due to safety issues, in spite of the desperate demand for insulin sensitizers [28,39]. Interestingly, PPARγ deacetylation is able to uncouple the adverse effects of TZDs, especially bone loss, from metabolic benefits. Here, we consistently showed that aKQ mice retained the same response to rosiglitazone, improving insulin sensitivity and glucose tolerance, but with a more pronounced bone loss in cotreatment with resveratrol. Besides its browning function [29,40], resveratrol has been shown to improve bone mineral density in postmenopausal women [41]. Hence, PPARγ deacetylation may be required to achieve this bone protection by resveratrol. The deleterious effect of PPARγ acetylation in adipocytes on the bone is very possibly mediated by Adipsin. This adipokine is a sensitive target to PPARγ acetylation and is increased in aKQ mice, while consistently repressed in 2KR mice. Adipsin can prime the bone marrow progenitor cells toward adipocyte differentiation, and the ablation of it alleviates bone loss in aging [21]. This aKQ model, thus, serves as a model of Adipsin gain-of-function to provide complementary evidence of its bone loss effect. Interventions targeting Adipsin and its related alternative complement pathways might convey bone protection, whereas exploiting its protection on insulin-producing β cells may have to be performed with the consideration of the potential adverse effects on the skeleton. 

Collectively, adipocyte PPARγ acetylation renders systemic metabolism toward obesity, insulin resistance, and dyslipidemia during aging. These metabolic dysfunctions are accompanied by the whitening of BAT, an adipocyte-intrinsic regulation by PPARγ acetylation (Figure 7). Targeting PPARγ acetylation might be promising for developing insulin sensitizers with improved safety to curb obesity, diabetes, and metabolic decline during aging. 

## Figures and Tables

**Figure 1 cells-12-01424-f001:**
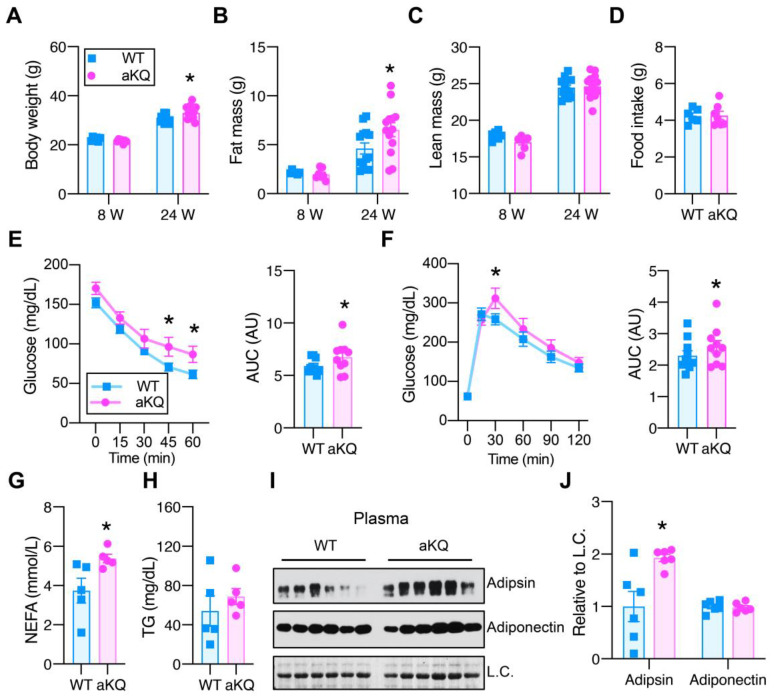
Adipose PPARγ acetylation impairs insulin sensitivity and glucose tolerance in middle-aged mice. (**A**–**C**) Body weight (**A**), fat mass (**B**), and lean mass (**C**) of male WT and aKQ mice at the age of 8 (n = 6 and 7) and 24 weeks (n = 13 and 13) on a chow diet. (**D**) Food intake of male aKQ and control mice at approximately 16 weeks of age. (**E**) ITT and area under curve (AUC) after fasting for 5 h in male WT and aKQ mice (at the age of 49 weeks) on a chow diet (n = 9 and 10). (**F**) GTT and AUC after an overnight fast in male WT and aKQ mice (at the age of 50 weeks) on a chow diet (n = 10 and 10). (**G**) Plasma non-esterified fatty acids (NEFAs) (n = 5 and 5) and (**H**) triglyceride (TG) levels in middle-aged mice on a chow diet (n = 5 and 5) after overnight fasting. (**I**,**J**) Adipsin protein levels determined by Western blotting (WB) with quantification in the plasma of middle-aged mice on a chow diet (n = 6, 6), signals were relative to loading control (L.C.). Data are presented as mean ± SEM, two-tailed Student’s *t*-test. * *p* < 0.05.

**Figure 2 cells-12-01424-f002:**
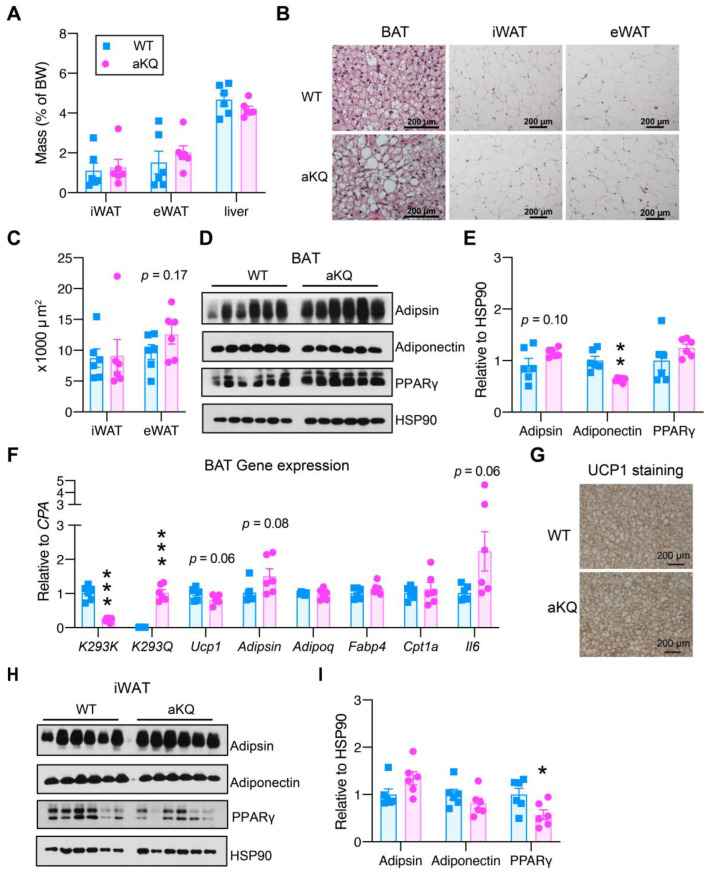
Differential effects on adipose tissues in middle-aged aKQ mice. Middle-aged male mice fed a chow diet were sacrificed after an overnight fast. (**A**) Tissue mass normalized to body weight. (**B**) Representative H & E staining of brown adipose tissue (BAT), inguinal WAT (iWAT), and epididymal WAT (eWAT). Scale bar: 200 µm. (**C**) Quantification of WAT adipocyte size. (**D**,**E**) WB with quantification of Adipsin, Adiponectin, and PPARγ proteins levels in BAT. (**F**) qPCR analysis of gene expression levels in BAT. (**G**) Immunohistochemical staining of UCP1 protein in BAT. (**H**,**I**) WB with quantification of Adipsin, Adiponectin, and PPARγ proteins levels in iWAT. Data are presented as mean ± SEM, two-tailed Student’s *t*-test. * *p* < 0.05, ** *p* < 0.01, *** *p* < 0.001, n = 6 and 6.

**Figure 3 cells-12-01424-f003:**
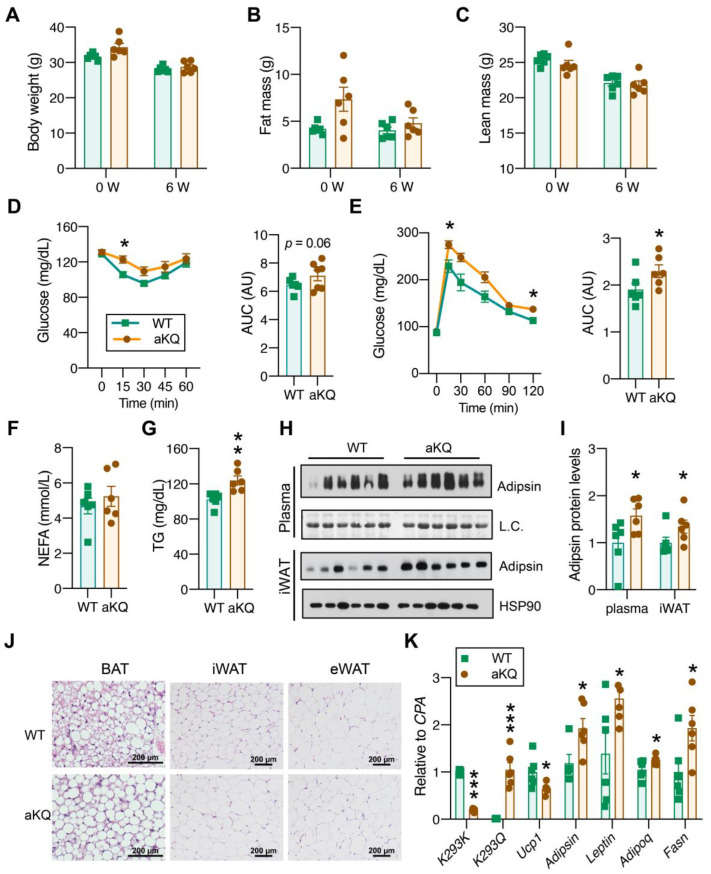
Adipocyte PPARγ acetylation diminishes the metabolic improvements of intermittent fasting (IF) in middle-aged mice. (**A**–**C**) Body weight (**A**), fat mass (**B**), lean mass (**C**) of male middle-aged WT and aKQ mice before and after 6 weeks of IF (n = 6, 6). (**D**) ITT and AUC after fasting for 5 h after 5 weeks of IF. (**E**) GTT and AUC after an overnight fasting after 6 weeks of IF (n = 6, 6). (**F**,**G**) Plasma NEFA and triglyceride (TG) levels after overnight fasting after 6 weeks of IF mice (n = 6, 6). (**H**,**I**) WB with quantification of Adipsin and Adiponectin protein levels in plasma and iWAT after 6 weeks of IF (n = 6, 6). (**J**) Representative H & E staining of BAT, iWAT, and eWAT after IF. Scale bar: 200 µm. (**K**) qPCR analysis of gene expression levels in BAT (n = 6, 6). Data are presented as mean ± SEM, two-way ANOVA and two-tailed Student’s *t*-test were used to test significances. * *p* < 0.05, ** *p* < 0.01, *** *p* < 0.001.

**Figure 4 cells-12-01424-f004:**
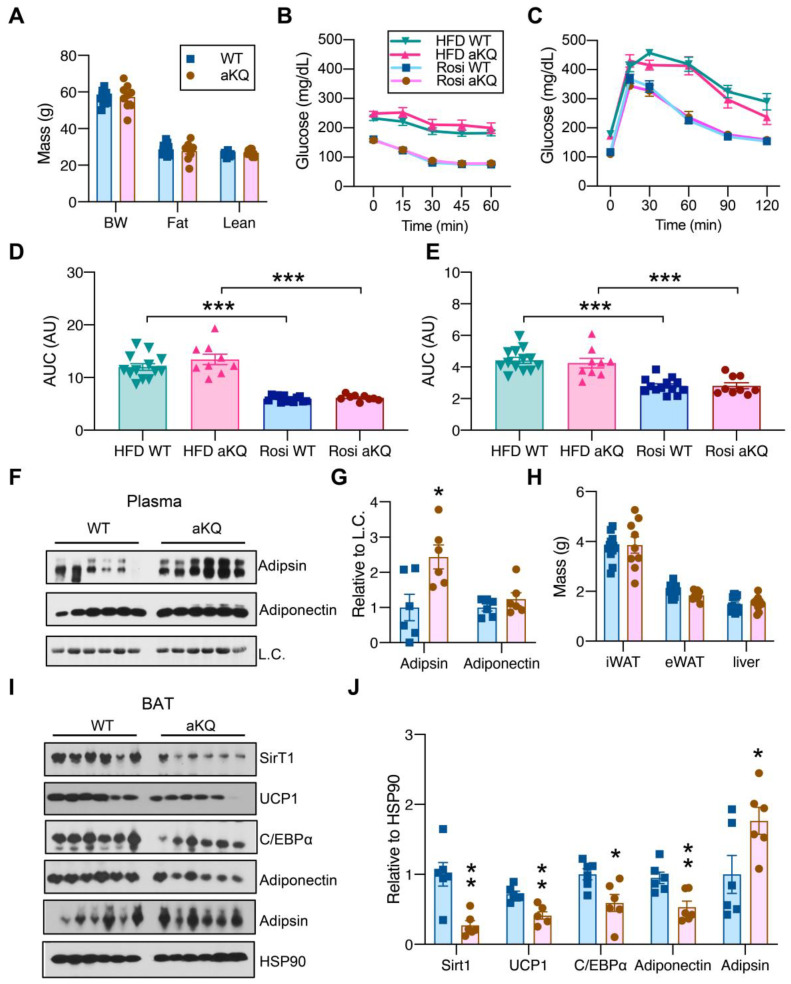
aKQ mice are sensitive to TZD treatment but with worsened BAT whitening. (**A**) Body compositions of diet-induced obese (DIO) WT and aKQ mice after a rosiglitazone (Rosi) diet for 7 weeks (n = 13, 9). (**B**). ITT on Rosi treatment for 0 and 5 weeks (n = 13, 9, 13, and 9). (**C**) GTT on Rosi treatment for 0 and 6 weeks (n = 13, 9, 13, and 9). (**D**) AUC of ITT (n = 13, 9, 13, and 9). (**E**) AUC of GTT (n = 13, 9, 13, and 9). (**F**,**G**) WB with quantification of Adipsin and Adiponectin protein levels in plasma (n = 6 and 6). (**H**) Tissue masses at sacrifice after 7 weeks of Rosi treatment (n = 13, 9). (**I**,**J**) WB with quantifications in BAT (n = 6 and 6). * *p* < 0.05, ** *p* < 0.01, *** *p* < 0.001.

**Figure 5 cells-12-01424-f005:**
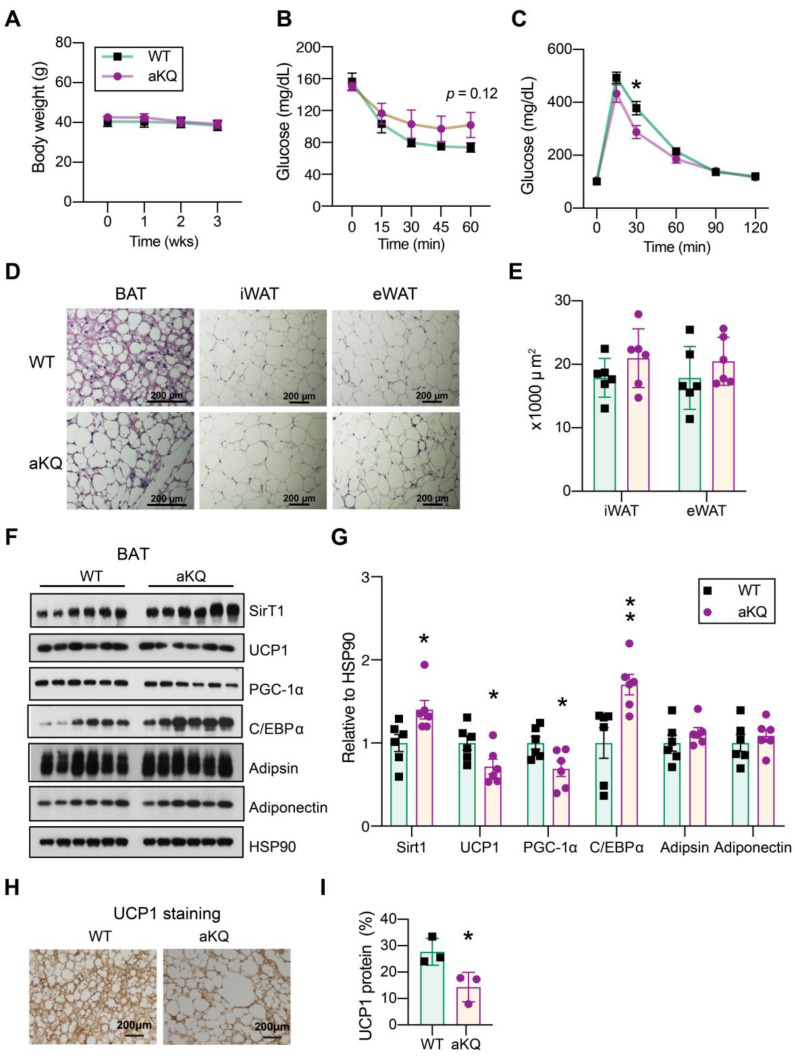
PPARγ acetylation-induced BAT whitening persists despite resveratrol treatment. (**A**) Body-weight curves of DIO mice co-treated with Rosi and resveratrol (n = 6, 6). (**B**) ITT and AUC after the co-treatment for 2 weeks (n = 6, 6). (**C**) GTT and AUC at 3 weeks co-treatment (n = 6, 6). (**D**). Representative H & E staining of BAT, iWAT, and eWAT after Rosi and resveratrol treatment for 3 weeks. Scale bar: 200 µm. (n = 6, 6). (**E**) Quantification of WAT adipocyte size (n = 6, 6). (**F**,**G**) WB and quantifications in BAT (n = 6, 6). (**H**,**I**) Immunohistochemical staining and quantification of UCP1 protein in BAT (n = 3, 3). Data ae presented as mean ± SEM, two-tailed Student’s *t*-test. * *p* < 0.05, ** *p* < 0.01.

**Figure 6 cells-12-01424-f006:**
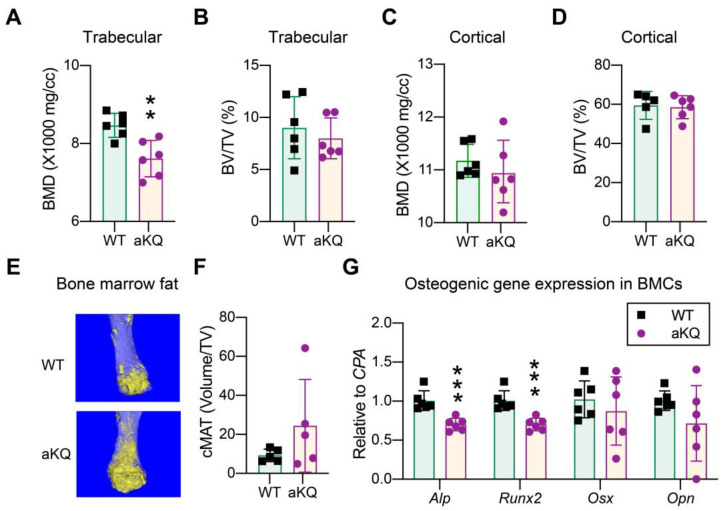
PPARγ acetylation accelerates TZD-induced bone loss in obese mice. (**A**) Bone mineral density (BMD) in the trabecular bone (n = 6, 6). (**B**) Normalized bone volume (BV) to total volume (TV) in the trabecular regions. (**C**) Bone mineral density in the cortical bone (n = 6, 6). (**D**) Normalized bone volume in the cortical regions (n = 6, 6). (**E**,**F**) Representative osmium tetroxide staining, and quantification of constitutive marrow adipose tissue (cMAT) assessed by μCT scanning in the femurs of male mice after Rosi plus resveratrol treatment (n = 5, 5). (**G**) qPCR analysis of osteoclastogenic genes (n = 6, 6). ** *p* < 0.01, *** *p* < 0.001.

**Figure 7 cells-12-01424-f007:**
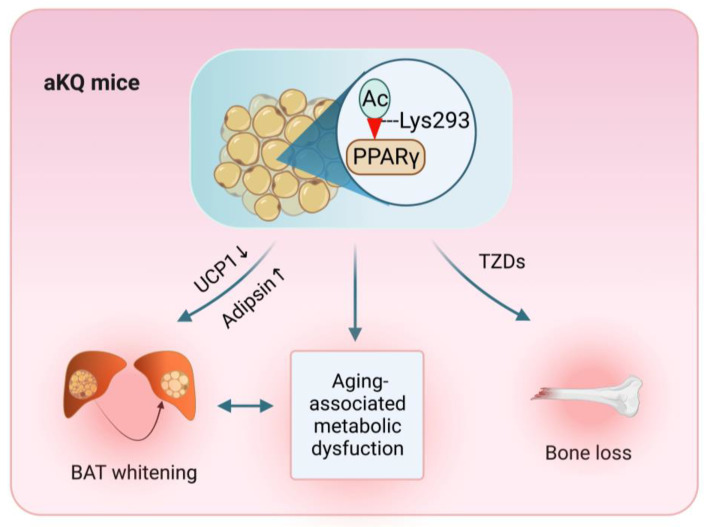
Schematic model of PPARγ acetylation in adipocytes exacerbating BAT whitening and worsening aging-associated metabolic dysfunction. The illustration was created using BioRender.

## Data Availability

Not applicable.

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
