# Peer review of "PPARγ Acetylation in Adipocytes Exacerbates BAT Whitening and Worsens Age-Associated Metabolic Dysfunction"

_cells, 2023, doi:10.3390/cells12101424_

Round 1
Reviewer 1 Report
Summary
In the manuscript by He et al., the authors report that acetylation of PPARγ triggers aging-associated metabolic dysfunction along with increased BAT whitening. They previously demonstrated that PPARγ acetylation regulates adipose plasticity and metabolic rhythms (PMID: 36394167). In this study, the authors show that constitutive acetylation of PPARγ (aKQ) mice have worsened aging-associated metabolic decline, accompanied by BAT whitening but not necessarily WAT. Additionally, aKQ mice fail to show any metabolic improvements from intermittent fasting (IF). The authors also demonstrate that TZD or Sirt1 fails to protect aKQ mice from BAT whitening. Furthermore, they reveal that PPARγ acetylation induces aging-dependent bone loss in obese mice. Therefore, they conclude that PPARγ acetylation in adipocytes could serve as a pathogenic factor predisposing mice to metabolic complications in aging. The experiments were well-designed, and the results were appropriately interpreted. The manuscript was clearly and logically written. No additional experiments are needed to substantiate their findings. However, the authors should consider the following suggestions to improve the clarity and accuracy of the manuscript:
1. While reporting the potential metabolic benefits of intermittent fasting (IF), the authors found an increased number of lipid droplets in aging IF BAT compared to aging control BAT (Figure 3J vs. Figure 2B). It would be helpful if the authors could clarify these interesting findings.
2. The authors demonstrated increased BAT whitening in aKQ mice. It would be helpful to know whether aKQ mice have more BAT tissue weight.
3. The authors claim that aKQ mice have normal iWAT and eWAT morphology and adipocyte size. It might be helpful if the size of adipocytes could be quantified.
4. The authors observed that some brown adipocytes in the brown adipose tissue (BAT) of aKQ mice appear unilocular, raising the question of whether these cells retain their brown identity or have characteristics of white adipocytes. Immunofluorescence staining for UCP1 could be a useful approach to investigate this matter.
5. It would be interesting to know whether aKQ mice consume similar amounts of food as their WT littermates.
6. In line 286, a reference is needed to support the statement being made.
7. In line 161, the authors should add "Figure 1H, I" to support the statement that adiponectin levels remain constant.
Author Response
Response:
We thank the reviewer for the insightful and constructive suggestions.
- While reporting the potential metabolic benefits of intermittent fasting (IF), the authors found an increased number of lipid droplets in aging IF BAT compared to aging control BAT (Figure 3J vs. Figure 2B). It would be helpful if the authors could clarify these interesting findings.
Response: We thank the reviewer for raising this interesting question. This finding has been clarified in the revised manuscript (Figure S1G). It is consistent with two previous studies. One is a mild mass increase in the interscapular BAT was observed in EODF mice (reference: Intermittent Fasting Promotes White Adipose Browning and Decreases Obesity by Shaping the Gut Microbiota. Cell Metab. 2017Oct3;26(4):672-685.e4.doi:10.1016/j.cmet.2017.08.019.); The other one is intermittent fasting also caused bigger lipid droplets size in that BAT of rats (Fasting and refeeding cycles alter subcutaneous white depot growth dynamics and the morphology of brown adipose tissue in female rats. Br J Nutr. 2021 Aug 14;126(3):460-469). Both references are included in the revised manuscript.
- The authors demonstrated increased BAT whitening in aKQ mice. It would be helpful to know whether aKQ mice have more BAT tissue weight.
Response: That is a good point. It is a pity that we did not measure BAT tissue weight because we couldn’t clearly tell the edge of BAT from the attached WAT at sacrifice.
- The authors claim that aKQ mice have normal iWAT and eWAT morphology and adipocyte size. It might be helpful if the size of adipocytes could be quantified.
Response: The adipocyte sizes of iWAT and eWAT have been quantified as shown in Figure 2C, Figure S1E, and Figure 5E.
- The authors observed that some brown adipocytes in the brown adipose tissue (BAT) of aKQ mice appear unilocular, raising the question of whether these cells retain their brown identity or have characteristics of white adipocytes. Immunofluorescence staining for UCP1 could be a useful approach to investigate this matter.
Response: That is an excellent point. We detected the UCP1 protein levels of BAT under three conditions including Aging, Aging IF and Rosi plus Res. It turns out that, UCP1 was significantly reduced in aKQ mice after Rosi plus Res treatment (Figure 5H and I), consistent with the UCP1 expression detected by western blot (Figure 5F and G). There was no significant change in Aging (Figure2G and Figure S1A) and Aging IF (Figure S1C) mice probably because of the low basal expression. These new data have been added to the revised manuscript.
- It would be interesting to know whether aKQ mice consume similar amounts of food as their WT littermates.
Response: The food intake of aKQ and WT are comparable on chow diet. We have added it as Figure 1D in the revised manuscript.
- In line 286, a reference is needed to support the statement being made.
Response: This has been corrected in the revised manuscript.
- In line 161, the authors should add "Figure 1H, I" to support the statement that adiponectin levels remain constant.
Response: This has been added to the revised manuscript.
Reviewer 2 Report
PPARγ activation by SirT1-mediated deacetylation protects animals from diet-induced obesity and insulin resistance. Using a unique adipocyte-specific PPARγ acetylation-mimetic mutant KI mouse (aKQ), the manuscript by He et al. demonstrated that PPARγ acetylation leads to metabolic dysfunction by promoting BAT whitening and bone loss. Collectively, these studies from the Qiang group establish PPARγ acetylation as a pathogenic factor for aging and obesity-related metabolic disorders. The results were clearly described and generally supported their conclusions. A few critics/recommendations are as follows:
1. 1-year is not technically aging in mice, they generally have a life-span of 2-3 years. It is recommended to use "middle age" to describe these animals.
2. Fat mass was increased in 24w aKQ mice (Fig. 1B), body weight was significantly higher when they are 1 year old (ref. 17), but WAT weights were similar between the groups (Fig. 2A). The discrepancy needs to be resolved or discusses. The authors could try to increase N number, combine the weight of all WAT depots, or calculate the WAT/BW percentage.
3. Does UCP1 protein reduce its expression in the BAT of aKQ mice? Do the aKQ demonstrate cold-intolerant or lipolysis defects?
4. 2-way ANOVA is required to test differences in Fig. 3A-C.
5. Does aKQ mutation have any negative effect on WAT browning, besides BAT whitening?
Author Response
We thank the reviewer for the insightful and constructive suggestions.
- 1-year is not technically aging in mice, they generally have a life-span of 2-3 years. It is recommended to use "middle age" to describe these animals.
Response: We thank the reviewer for this important suggestion. This has been changed in the revised manuscript. We also changed to title “aging-associated metabolic dysfunction” to “age-associated metabolic dysfunction” in responding to reviewer’s point.
- Fat mass was increased in 24w aKQ mice (Fig. 1B), body weight was significantly higher when they are 1 year old (ref. 17), but WAT weights were similar between the groups (Fig. 2A). The discrepancy needs to be resolved or discusses. The authors could try to increase N number, combine the weight of all WAT depots, or calculate the WAT/BW percentage.
Response: We have tried but failed to find any significant increase in fat depot size. We think the reason is that the mice were sacrificed after overnight fasting and thus the modest increase in fat mass was blunted.
- Does UCP1 protein reduce its expression in the BAT of aKQ mice? Do the aKQ demonstrate cold-intolerant or lipolysis defects?
Response: UCP1 proteins were not significantly decreased in aging and aging-IF aKQ mice by Immunohistochemical staining (Figure 2G, Figure S1A, and Figure S1C). After treatment with Rosi or Rosi plus Res, UCP1 proteins were significantly repressed by western blotting (Figure 4J-K, 5F-G) and Immunohistochemical staining (Figure 5H, I). In aging IF mice, aKQ mice showed a repressed trend in lipolysis. This data has been added to the revised manuscript (Figure S1D).
- 2-way ANOVA is required to test differences in Fig. 3A-C.
Response: This has been corrected in the revised manuscript.
5. Does aKQ mutation have any negative effect on WAT browning, besides BAT whitening?
Response: We did not detect significant changes in brown markers Ucp1, Dio2 and Cidea in WAT (Figure S1B and Figure 1F), probably because WAT browning was minimal under these conditions and aKQ mice can’t reduce it further.
Reviewer 3 Report
This manuscript by He et al. describes the contribution of hyperacetylated PPARgamma on age-related metabolic dysfunction, especially the whitening of BAT. Moreover, the authors also showed that hyperacetylated PPARgamma is also responsible for bone loss, one of the adverse effects of PPARgamma agonists. The way the authors explore the topic is interesting, but some issues should be addressed before publication.
1. Is aKQ mice responsive to rosiglitazone? The authors show in Figure 4 that insulin sensitivity and glucose tolerance were similar between WT and aKQ. However, they lack some control data, the ones at times before feeding HFD and when starting rosiglitazone treatment.
2. To my understanding, TZDs induce browning (Qiang et al., Cell 150 620–632, 2012). If I read it right, the reference the authors cited (Ref. #27) does not mean that TZDs induce fat accumulation in BAT. It’s a bit confusing.
3. Probably authors meant neutral buffered formalin solution on page 3, line 114.
Author Response
We thank the reviewer for these insightful and constructive suggestions.
- Is aKQ mice responsive to rosiglitazone? The authors show in Figure 4 that insulin sensitivity and glucose tolerance were similar between WT and aKQ. However, they lack some control data, the ones at times before feeding HFD and when starting rosiglitazone treatment.
Response: Yes, the aKQ mice responded similarly to rosiglitazone as WT. We have included ITT and GTT before Rosi treatment to Fig. 4B and 4C, respectively.
- To my understanding, TZDs induce browning (Qiang et al., Cell 150 620–632, 2012). If I read it right, the reference the authors cited (Ref. #27) does not mean that TZDs induce fat accumulation in BAT. It’s a bit confusing.
Response: We thank the reviewer for pointing out this mistake. It has been corrected in the revised manuscript.
- Probably authors meant neutral buffered formalin solution on page 3, line 114.
Response: Thanks! We corrected this mistake.
Round 2
Reviewer 3 Report
All issues have been addressed.